# Myopia and Near Work: A Systematic Review and Meta-Analysis

**DOI:** 10.3390/ijerph20010875

**Published:** 2023-01-03

**Authors:** Frédéric Dutheil, Tharwa Oueslati, Louis Delamarre, Joris Castanon, Caroline Maurin, Frédéric Chiambaretta, Julien S. Baker, Ukadike C. Ugbolue, Marek Zak, Ines Lakbar, Bruno Pereira, Valentin Navel

**Affiliations:** 1Preventive and Occupational Medicine, University Hospital of Clermont-Ferrand, Physiological and Psychosocial Stress, CNRS, LaPSCo, Université Clermont Auvergne, WittyFit, 63001 Clermont-Ferrand, France; 2Ophthalmology, University Hospital of Clermont-Ferrand (CHU), 63001 Clermont-Ferrand, France; 3Department of Anesthesiology and Intensive Care, University Hospital of Marseille, Hopital Nord—Assistance Publique Hôpitaux de Marseille, 13015 Marseille, France; 4Centre for Health and Exercise Science Research, Physical Education and Health, Kowloon Tong, Hong Kong Baptist University, Hong Kong; 5School of Health and Life Sciences, University of the West of Scotland, Glasgow G72 0LH, UK; 6Faculty of Medicine and Health Sciences, Institute of Physiotherapy, The Jan Kochanowski University, 25-369 Kielce, Poland; 7Biostatitics, Clinical Research Direction, University Hospital of Clermont-Ferrand (CHU), 63001 Clermont-Ferrand, France

**Keywords:** near work, myopia, occupation, epidemiology, public health

## Abstract

Background: Myopia is a global public health problem affecting quality of life and work productivity. Data is scarce regarding the effects of near work on myopia. Providing a larger meta-analysis with life-long perspective, including adults and occupational exposure seemed needed. Methods: We searched PubMed, Cochrane Library, Embase and Science Direct for studies reporting myopia prevalence in near work. Myopia was defined as a mean spherical equivalent ≤ −0.50 diopter. We performed a meta-analysis using random-effects model on myopia prevalence, myopia progression per year, and odds ratio (OR) of myopia in near work, completed by subgroup analyses and meta-regressions on patients’ characteristics, type of work in adults, geographic zones, time and characteristics of near work. Results: We included 78 studies, representing a total of 254,037 participants, aged from 6 to 39 years. The global prevalence of myopia in near work was 35% (95% CI: 30 to 41%), with a prevalence of 31% (95% CI: 26 to 37%) in children and 46% (95% CI: 30 to 62%) in adults. Myopia progression was −0.39 diopters per year (−0.53 to −0.24 D/year), ranging from −0.44 (−0.57 to −0.31) in children to −0.25 D/year (−0.56 to 0.06) in adults. The odds of myopia in workers exposed vs. non-exposed to near work were increased by 26% (18 to 34%), by 31% (21 to 42%) in children and 21% (6 to 35%) in adults. Prevalence of myopia was higher in adults compared to children (Coefficient 0.15, 95% CI: 0.03 to 0.27). Conclusions: Near work conditions, including occupational exposure in adults, could be associated with myopia. Targeted prevention should be implemented in the workplace.

## 1. Introduction

Myopia is the most common type of refractive error [1] and constitutes a rising global health issue causing significant impact on visual function [2,3] with potential consequences on quality of life and work productivity [4,5] and other sequelae (glaucoma, macular degeneration, retinal detachment and cataract) [6,7]. Myopia most often involves the axial elongation of the eyeball [8,9], and can be caused by genetic [10,11] and environmental factors [12,13]. The increasing worldwide prevalence of myopia is thought to be caused by the progression of its environmental factors [14,15]. Myopia can begin at any age but several studies report the association of increased myopia prevalence with higher education and occupational status [16,17], and in highly industrialized countries [12]. Near work is defined as a work involving proximity of the eyes to an object requiring accommodation such as paper reading or computer work using a Visual Display Terminal (VDT). Myopia progression is associated with long-term changes in the musculo-fascial system [18] and could be associated with prolonged near work and less time spent performing sports and outdoor activities among children and adults [19]. Despite an increasing body of research, results remain conflicting. The only meta-analysis regarding the incidence of myopia [20] was limited to children and did not report myopia progression across time. Providing a larger life-long perspective and comprehensive meta-analysis including adults and occupational exposure seemed therefore needed.

The primary aim of the present study was to conduct a systematic review and meta-analysis to determine the global prevalence of myopia relative to occupation (including education). The secondary outcomes were to assess the effects of near work on myopia, through a meta-analysis on myopia progression and odds ratio of myopia in exposed patients to near work vs. those not exposed to near work, to determine the dose-response relationships between myopia and near work and to examine the influence of the occupation-related near work and socio-demographic variables.

## 2. Materials and Methods

### 2.1. PICO Question

The Population-Intervention-Comparison-Outcome (PICO) question was formulated as follows: Do adults and children (P) exposed to near work either from their occupation or their educational activities (I) compared to adults and children that are not exposed to near work (C) have a different prevalence of myopia (O)? 

### 2.2. Literature Search

The present research was carried out in accordance with the PRISMA Guidelines. We reviewed all cohort studies involving myopia in near work. Specifically, the inclusion criteria for the search strategy were cross-sectional and cohort studies (minimal number of 10 individuals), without a case-study design. We used the following keywords: “myopia” or “short-sight” for myopia, and “near work”, “display” or “users” for near work. A display is hereafter defined as a computer output surface and projecting mechanism that shows text and images onto a screen, using a cathode ray tube, liquid crystal display, light-emitting diode, gas plasma, or other image projection technology. To be included, articles needed to describe the outcomes of interest of the present research, i.e., the number of workers with myopia, myopia progression, or odds ratio for myopia. The following databases were searched up until 30 May 2022: PubMed, Cochrane Library, Science Direct and Embase. The search was not limited to a specific time period. In addition, reference lists of all publications meeting the inclusion criteria were manually searched to identify any further studies not found through electronic research on the databases stated above. Articles written in Chinese or Japanese language were excluded. The search strategy is described in Appendix A. Two authors (J. Castanon and T. Oueslati) conducted all literature searches, collected the abstracts, separately reviewed the abstracts and rated the suitability of the articles for inclusion, based on the criteria mentioned above. Another author (F. Dutheil) reviewed articles for which consensus on suitability was not met by the first 2 raters. 

### 2.3. Quality Assessment

We used the Scottish Intercollegiate Guidelines Network (SIGN) for cohort studies to evaluate the risk of bias of the included articles [21]. Not designed for quantifying the internal validity of studies [22], the “STrengthening the Reporting of OBservational studies in Epidemiology” (STROBE) criteria [23] were also used to evaluate the quality of reporting. For both checklists, one point was attributed per completed item, then converted into a percentage to give a score of risk of bias and reporting quality for each included study (Appendix A). 

### 2.4. Data Collection

The collected data included first author’s name, publication year, study design, country, aims and outcomes, sample size, characteristics of individuals (age, sex, education level, parental history of myopia, occupation for adults), myopia outcomes (prevalence of myopia, myopia progression, i.e., diopter changes per year, risk of myopia or odds ratio of myopia), use of cycloplegia or not for the diagnosis of myopia and characteristics of near work (computer, paper reading, indoor and outdoor activities, diopter-hour per week, visual display terminal exposure).

### 2.5. Statistical Considerations

Baseline characteristics were summarized for each study and reported as mean ± standard-deviation (SD) for continuous variables, and number (%) for categorical variables. Due to the observational nature of the included studies, we chose to using random-effect models for the present meta-analysis (DerSimonian and Laird approach) [24,25] to assess myopia prevalence, myopia progression, and risk of myopia following exposure to near work, with results expressed as percentage, diopter change per year, and odds ratio with their 95% confidence interval (95% CI), respectively. We conducted subgroup analysis by age (adults above 18 years old vs. children/teenagers), type of work in adults, continents, and time periods (before 2005, between 2005 and 2015, and after 2015). Statistical heterogeneity between results was assessed by examining forest plots, confidence intervals and I-squared (I^2^). Heterogeneity was considered low for I^2^ < 25%, modest for 25 < I^2^ < 50%, and high for I^2^ > 50%. The robustness of our results was assessed by conducting a sensitivity analysis after exclusion of studies that were not evenly distributed around the funnel plot. When sample size was sufficient, meta-regressions were used to study the relationships between our outcome variables (prevalence, myopia progression, and risk for myopia) and clinically relevant parameters such as characteristics of individuals (age, sex, education level, parental history of myopia), type of occupation for adults, continents, time periods (before year 2005, years 2005 to 2015, after 2015), duration and characteristics of near work (computer, paper reading, indoor and outdoor activity, diopter-hour per week, VDT exposure). Results were expressed as regression coefficients and 95% CI. All *p* values were two-tailed. Statistical significance was defined by a *p* value < 0.05. Statistical analysis was conducted using Stata software (v16, StataCorp, College Station, TX, USA).

## 3. Results

An initial search produced retrieved 9617 articles (Figure 1). Removal of duplicates and application of selection criteria reduced the number of articles reporting myopia in near work to 78 studies [2,13,14,15,19,26,27,28,29,30,31,32,33,34,34,35,36,37,38,39,40,41,42,43,44,45,46,47,48,49,50,51,52,53,54,55,56,57,58,59,60,61,62,63,64,65,66,67,68,69,70,71,72,73,74,75,76,77,78,79,80,81,82,83,84,85,86,87,88,89,90,91,92,93,94,95,96]. All studies were written in English except one in French [85] and one in Italian [30]. Main characteristics of the studies are presented below and summarized in Table 1.

### 3.1. Quality of Articles

Using the SIGN checklist, overall quality of the 78 included studies was good, with a mean score of 72.2 ± 8.5%, ranging from 40.0 [12,85] to 86.0% [69]. The main limitation was a confusion bias with few studies considering parental history of myopia. For the studies reporting myopia progression, the main bias was the number of dropouts that was seldom reported (Figure 2 and Appendix A). The use of the STROBE checklist confirmed the overall good reporting quality of the included articles with a mean score of 81.2 ± 11.1%, ranging from 54.7 [57] to 96,7% [69,99]. All articles either did not report the description of confounding factors or did not consider the use of a flow chart. Only half of studies reported an ethical approval.

### 3.2. Population

Population sizes ranged from 40 [32,34,35,43,54,55,56,59,65,67,68,71,76] to 23,314 [61]. In total, 254,037 individuals involved in near work were included in this review. *Age* of participants ranged from 6 [41] to 39 years [40], with 19 studies being on adults and 58 in children/teenagers. Seven of the eight studies that did not report age were on children [28,34,44,58,61,66,71,85], and one in adults [85]. Proportion of *male* ranged from 24% [85] to 100% [15,32,80,83,91] and was not specified in nine studies [38,44,45,52,53,58,70,87,92]. Twenty two studies reported the prevalence of myopia separately for each gender [2,13,14,32,34,35,43,55,56,59,65,67,68,71,76,77,81,84,88,89,90]. The socio-professional category of adults was specified in all studies: administrative workers [30,85], military conscripts [32,57], university students [28,31,40,50,52,74,76,78,87,88,97], and even fishermen [2]. All the studies reported the education level except 29 studies [13,14,19,35,36,37,41,43,44,47,50,54,55,58,59,61,62,63,64,68,71,72,75,81,82,88,93,94,99]. All the studies reported the parental history of myopia except 18 studies [2,14,19,30,32,36,37,40,47,52,59,66,72,74,85,87,88]. Geographic zones of studies were North America [14,33,35,53,67,75,90,94], Oceania [41,70,81,87,90,94], Europe [13,19,26,38,39,40,42,47,52,57,71,72,73,74,76,77,79,85,97], and Asia [2,15,34,36,37,43,44,45,46,48,49,50,51,56,58,59,59,60,61,61,62,63,66,68,69,78,80,83,84,86,88,89,91,93,95,95,99]. 

### 3.3. Study Designs and Outcomes of Included Studies

The main outcome was the *prevalence* of myopia in 61 studies with a cross-sectional design [2,14,15,26,27,28,29,30,31,32,33,34,36,39,42,43,44,46,48,48,50,51,55,57,60,62,63,64,65,66,67,68,70,71,73,75,77,78,79,81,83,84,86,87,88,89,90,92,94] and 16 studies aimed at *relating myopia* (autorefractometry, axial length) *with near work exposure* (kind of near work, and duration) in prospective case-control designs [12,13,35,38,40,41,45,47,51,52,53,54,58,59,61,69,69,72,76,80,82,85,87,91,93,95,97]. The diagnosis criteria for myopia retained was ≤ −0.50 Diopter (D). 

*Myopia prevalence* was described in 72 studies [2,12,13,14,15,26,27,28,29,30,31,32,33,34,35,36,37,38,39,40,41,42,43,45,46,47,48,49,50,51,52,53,54,55,56,57,58,59,60,61,62,63,65,66,67,68,70,71,72,73,74,75,76,77,78,79,81,82,83,84,85,86,87,88,90,92,94,95,95,97], *myopia progression* per year in 17 studies [13,34,36,37,45,51,52,54,59,72,74,76,80,90,93,95,97] and *odds ratio for myopia* in 22 studies [2,30,31,32,34,38,40,41,42,46,48,49,51,56,60,61,62,64,65,67,68,69,70,75,76,77,80,81,83,84,86,88,89,93]. Thirty studies combined two or three outcomes (myopia prevalence, and/or myopia progression, and/or odds ratio) [1,2,3,4,8,9,11,12,13,18,19,20,21,28,29,30,31,32,33,34,35,36,37,38,39,40,41,42,43,44,46,47,48,49,51,53,54,59,61,62,66,67,68,69,71]. Within the 17 studies reporting the myopia progression, only two did not report the prevalence of myopia [80,93]. The number of studies in each occupation remained low.

### 3.4. Methods to Measure Myopia

All studies used *autorefractometry* for the diagnosis of myopia aiming to calculate the refractive error based on spherical equivalent (SE = sphere + cylinder/2). Most studies used cycloplegia, except 16 studies, mostly on adults [15,31,32,46,47,49,50,55,56,66,67,71,73,78,79,88]. Seventeen studies quantified *axial length* [15,38,39,44,59,60,64,79,83,84,87,90,92,93,94], and two studies reported perceptions of visual fatigue [26,97]. Studies evaluating myopia progression considered a pathologic refractive error for a SE change of −0.50 D [8,9,11,13,19,20,29,31,35,36,40,42,44,45]. Eleven studies reported the prevalence of high myopia defined by at least −6 D [15,32,43,46,51,56,78,83,84,88,92]. 

### 3.5. Quantification of Exposure to Near Work

Twenty-six studies reported the *duration of exposure* to near work, ranging from few weeks [67] to 23 years [13]. 

Twenty studies [12,14,15,29,30,35,40,43,53,57,58,62,63,68,70,74,75,81,85,95] reported a weekly use of computer work, ranging from 0 [57] to 45 h [40]. Twenty-six studies reported the weekly duration of reading papers [13,14,15,19,27,29,35,43,45,53,57,58,60,62,63,68,70,71,74,75,76,78,84,88,95,97] ranging from 2.8 [84] to 33 h [63]. Similarly, 25 studies [14,15,29,35,39,40,41,42,43,44,45,53,61,62,63,68,69,75,79,81,82,83,92,93,95] reported a weekly duration of outdoor activity ranging from 4.4 [82] to 21 h [13]. Seven studies [29,39,43,45,62,73,78] reported a weekly duration of use of electronic devices ranging from 2.80 h [62] to 30 h [74]. 

Only 18 studies reported the global week duration of near work [28,29,33,37,42,49,50,53,54,68,75,76,76,81,84,85,86,90] ranging from 10 [64] to 65 h [83]. With the exception of eight studies [14,37,63,65,67,68,90,92], all studies reported an association between the prevalence of myopia and the duration of computer work, paper reading and decreased outdoor activity. 

Seven studies quantified the exposure to near work in Diopter-hours per week [26,41,53,74,75,84,95] ranging from 20 [26] to 134 [75]. The duration of the different kinds of near work was estimated by questionnaire. Four studies attributed a specific accommodative effort of 3D to reading or writing papers and 2D for computer work [14,41,74,75]. Then, the accommodative effort was multiplied by the duration of the activity in hours i.e., 3 h of computer work per day during 5 days equals to 30 Diopter-hours per week. Two studies did not specify the measurement of Diopter-hours [59,75] and one study quantified Diopter-hours as the duration of near work multiplied by the reciprocal of the distance from which the activity was performed [82]. All the studies reporting Diopter-hours per week also reported the prevalence of myopia except one [53]. 

### 3.6. Meta-Analysis on Prevalence of Myopia

Within the 74 studies reporting a prevalence [1,2,3,4,8,9,11,12,13,18,19,20,21,28,29,30,31,32,33,34,35,36,37,38,39,40,41,42,43,44,47,48,49,51,52,53,54,59,62,66,67,68,69,70,71], the overall prevalence of myopia was 35% (95% CI: 30 to 41%), with heterogeneous results ranging from 1.5% [81] to 88.1% [83]. Stratification by age demonstrated a prevalence of 46% for adults (95% CI: 30 to 62%) and a prevalence of 32% for children (95% CI: 26 to 37%). Occupations were heterogeneous. University students constituted the only group of adults for whom more than three studies were available, with a prevalence of myopia estimated to 36% (95% CI: 19 to 55%). Stratification by geographic zones demonstrated a prevalence of 34% (95% CI: 26 to 43%) in North America, 18% (95% CI: 9 to 29%) in Oceania, 24% (95% CI: 19 to 30%) in Europe, 27% (95% CI: 13 to 44%) in the Middle East, and 48% (95% CI: 40 to 56) in Asia. The prevalence ranged from 29 to 47% depending on stratification by time. Heterogeneity was high in all sensitivity analysis (I^2^ > 90%). Those results are presented in Figure 3. Sensitivity analysis yielded similar results. 

### 3.7. Meta-Analysis on Myopia Progression Per Year

Within the 17 studies reporting a myopia progression [8,9,11,13,19,20,29,31,35,36,40,42,44,45,47,49,53,55,57,60,61], the global myopia progression was −0.39 D per year (95% CI: −0.53 to −0.24, *p* < 0.001), with heterogeneous results ranging from −0.05 [13] to −0.92 D per year [45]. After stratification by age, myopia progression was −0.25 D per year in adults (−0.56 to 0.06, *p* = 0.119) and −0.44 D per year in children (−0.57 to −0.31, *p* < 0.001). Insufficient data precluded analysis by occupation. Stratification by geographic zones demonstrated a significant myopia progression of −0.48 D per year for Asians (−0.55 to −0.41, *p* < 0.001) but not in the other continents. Stratification by time demonstrated a myopia progression of −0.38 D per year (−0.54 to −0.21 D/year) before 2005, −0.26 D per year (−0.36 to −0.16 D/year) between 2005 and 2015, and −0.47 D per year (−0.54 to −0.40 D) after 2015 (*p* < 0.001). All I^2^ were > 90%. These results are presented in Figure 4.

### 3.8. Meta-Analysis on Odds Ratio

Within the 22 studies estimating an OR for myopia when exposed to near work [1,2,3,9,18,20,31,32,33,37,38,39,40,41,48,52,53,54,57,59,60,61,62,67,68], the global exposure to near work increased the odds of myopia by 26% (odds ratio = 1.26, 95% CI: 1.18 to 1.34), with heterogeneous odds ratio ranging from 0.73 [6] to 9.30 [32]. More specifically, the odds of myopia in case of exposure to near work are increased by 21% for adults (OR = 1.21, 95% CI: 1.06 to 1.34) and by 31% for children (OR = 1.31, 95% CI: 1.21 to 1.42). Occupations were heterogeneous. Only university students were studied, with odds ratios provided in more than one study, with odds of myopia 22% higher in the group exposed to near work (1.22, 95% CI: 1.07 to 1.36). Stratification by geographic zones demonstrated increased odds of myopia in the population exposed to near work vs. the controls, by 21% in Asia (1.21, 95% CI: 1.09 to 1.32), and by 15% in Europe (1.15, 95% CI: 1.07 to 1.23). The increased odds of myopia in the population exposed to near work vs. controls remained stable across time. All *p*-values were < 0.001 and all I^2^ were > 75% (Figure 5). 

### 3.9. Meta-Regressions

There was a greater prevalence of myopia in adults than in children (coefficient 0.15, 95% CI: 0.03 to 0.27), in women compared to men (0.60, 95% CI: 0.30 to 0.90), and in Asia compared to Europe (0.21, 95% CI: 0.10 to 0.33) and Oceania (0.28, 95% CI: 0.09 to 0.47) (*p* < 0.001). There was a trend for a greater prevalence of myopia with the increase of indoor activity (0.11, −0.03 to 0.24, *p* = 0.090). Except a trend for a decreased prevalence of myopia in recent years, we did not find any other variables linked with prevalence of myopia (Figure 6). 

Regarding the progression of myopia, there was a trend for a greater myopia progression in Asia compared with North America (0.25, −0.13 to 0.63, *p* = 0.095) (Appendix A). We did not find any other variables linked with myopia progression, as well as no significant variables influencing the odds of myopia (Appendix A).

## 4. Discussion

The main findings were that near work exposure, including occupational exposure in adults, could be associated with myopia. The odds of myopia in the population exposed to near work is increased by 31% in children, and by 21% in adults. Some regions such as Asia may be more susceptible to the risk of myopia in case of near work exposure. We then discussed the factors that may contribute to this association between the risk of myopia and near work exposure.

### 4.1. Myopia and Near Work: Also in Adults

This study is the first meta-analysis that studied myopia and near work in adults. We demonstrated that occupational exposure to near work in adults is associated with a 20% increase in the odds of myopia. The increasing prevalence of myopia also implies untreatable complications that might occur even in the working-age population, such as myopic maculopathy, more frequently observed in high myopia [7]. In addition to its impact on visual function [1] myopia can lead to fatigue, anxiety [30,85] and distress at work [16], affecting the patient’s quality of life. Occupations at risk of myopia are those with an important accommodative effort [100] such as clinical microscopy [26], law student [67] or medical students [76], among others. These studies suggest that near work might participate to myopia onset and progression in young adults, but the mechanisms at stake are not clearly understood. The data regarding accommodative effort by occupation is scarce in the literature, therefore limiting the possible explorations of the relationship between near work and myopia in the workplace. While we regret that the comparisons between the prevalence of myopia and types of occupation were not feasible in our meta-analysis, myopia as a public health concern seems overlooked, despite economic implications both for employees and employers [101,102,103,104,105]. Therefore, preventative actions at the workplace should target screening for myopia and may influence management strategies [101]. The literature being very scarce in terms of studies regarding myopia in adults and its risk factors, prospective longitudinal studies in adult on the matter are needed. 

### 4.2. Effect of Time and Influence of Geographic Zones

Myopia prevalence in our study was similar to those previously reported globally [1] and in Europe [16]. Our results did not support the suspected increased prevalence of myopia. However, nearly all included studies were conducted after 2000, whereas the increased prevalence was suggested by comparing prevalence of myopia between 1930 and now [16]. Moreover, diagnosis of myopia during the 20th century may have become more and more systematic, potentially explaining this increased prevalence. Our meta-analysis highlighted heterogeneous results, with wide variations of the prevalence of myopia from 1.5% [83] to 88.1 [81]. Our results in the regional myopia prevalence (USA 29%, Europe 24%, Asia 48 %, Oceania 18%) are in accordance with those reported in the literature [106]. The role of genetic factors in geographic variations have been discussed, especially in East-Asia [101]. Nevertheless, a large number of genes found to be potentially linked with myopia with low penetrance [107,108] suggests a common predisposition towards myopia in the environments with higher amount of near work and lower outdoor light exposure [109,110,111,112], a feature especially frequent in Asia.

### 4.3. Other Factors Influencing Myopia

Factors promoting myopia are also specific to some cultural or subcultural characteristics, such as urbanization, economy, social class, education and lifestyle [19,37,41,113]. Despite potential additive effects of education and age were suggested as increasing the risk of myopia [16], the link between these factors (reading, age and education) makes assessment of separate effects difficult [19,57,97]. The presence of myopia in one parent multiplies by two the risk for a child to be myopic [114]. In our meta-analysis, the adult patients were rather young—mostly under 40 years old. Myopia seems to be more present in young adults, and seems to decrease secondarily [16] due to physiological aging or to the onset of a corticonuclear or cortical cataract [115]. The duration of the post-secondary school education could be in itself a risk factor of myopia more than the cognitive performance [116]. On the contrary, despite regular outdoor activities have been shown to have a beneficial effect in reducing the onset of myopia [117], our data could not confirm this result.

### 4.4. Myopia Progression in Near Work and Dose Response Relationships 

Our meta-analyses suggest that myopia progression may be associated with the exposure to near work [8,9,11,13,19,20,29,31,35,36,40,42,44,45]. Despite being vaguely defined in the literature, indoor activities are generally considered as a risk factor for myopia [41,42]. Our results confirm this association. Although myopia progression is generally linked with an increasing accommodative effort [9,20,32,39], we did not retrieve significant differences between paper reading (accommodation at 50 cm) and computer work (accommodation at 80 cm). To our knowledge, no previous studies have specifically compared the influence of paper reading and computer work on myopia. According to some studies, myopia is significatively associated with continuous reading, longer duration of reading and shorter reading distance [51,70,118], due to the promotion of eye growth by sustained accommodation. Moreover, myopia may also be linked to the issue of contrast between support and text, as suggested by Aleman et al. [119] and Wang et al. [120]. One controversial hypothesis is that computer work might prevent myopia thanks to the release of dopamine induced by the brightness of digital screens, which could prevent eye growth (i.e., axial length), and therefore myopia [121]. Taking breaks after 30 min of reading may protect against myopia [50], leading to innovative device developed to monitor near work, triggering an alert if a risky behaviour is detected [122]. 

### 4.5. Limitations

Our study has some limitations. Several biases could have been introduced via the literature search and selection procedure. We conducted the meta-analyses on published articles only; thus potentially exposed to publication bias. We observed that a large part of the variability in out meta-analysis is secondary to between-studies heterogeneity. Conclusions from subgroup analysis and meta-regressions were also affected by this high heterogeneity. The populations investigated in our meta-analysis appear to be from varied origin, predominantly from the developed and developing countries, favouring the external validity of our results toward the population exposed to near work. One potential confounding factor could be the amount of exposure to outdoor light one is exposed to, a parameter not reported in every included study. We could hypothesize that outdoor light exposure may be inversely correlated with indoor light exposure and near work, and thus could also be associated with myopia, but our results do not support this hypothesis (Figure 6). Moreover, meta-analyses are designed to take into account heterogeneous conditions [123]. Non-significant relationships between myopia and characteristics of population might be due to an insufficient sample size and other confounding factors. All included studies used an autorefractometer to measure the degree of myopia, however, some studies did not use cycloplegia which can induce detection bias. Finally, within our meta-analysis, most studies specified computer work as the VDT use. However, VDT may now relate to various screen distance (such as desktop computer, laptop, tablet, or smartphone) and distance is key factor of accommodation. Further studies should describe more precisely the distance of VDT exposure to allow for more precise estimation of the role of distance between the eye and object of focus in near work.

## 5. Conclusions

In conclusion, near work seems to be associated with myopia, including occupational exposure in adults. The prevalence of myopia following near-work exposure was significantly higher in adults (46%) compared to children (31%). Furthermore, our results suggest that the odds of myopia in adults with occupational exposure to near work is 21% higher (95% CI: 6 to 35%) compared to the non-exposed. Considering the consequences of myopia both for employees and employers, targeted prevention strategies should be implemented in the workplace.

## Figures and Tables

**Figure 1 ijerph-20-00875-f001:**
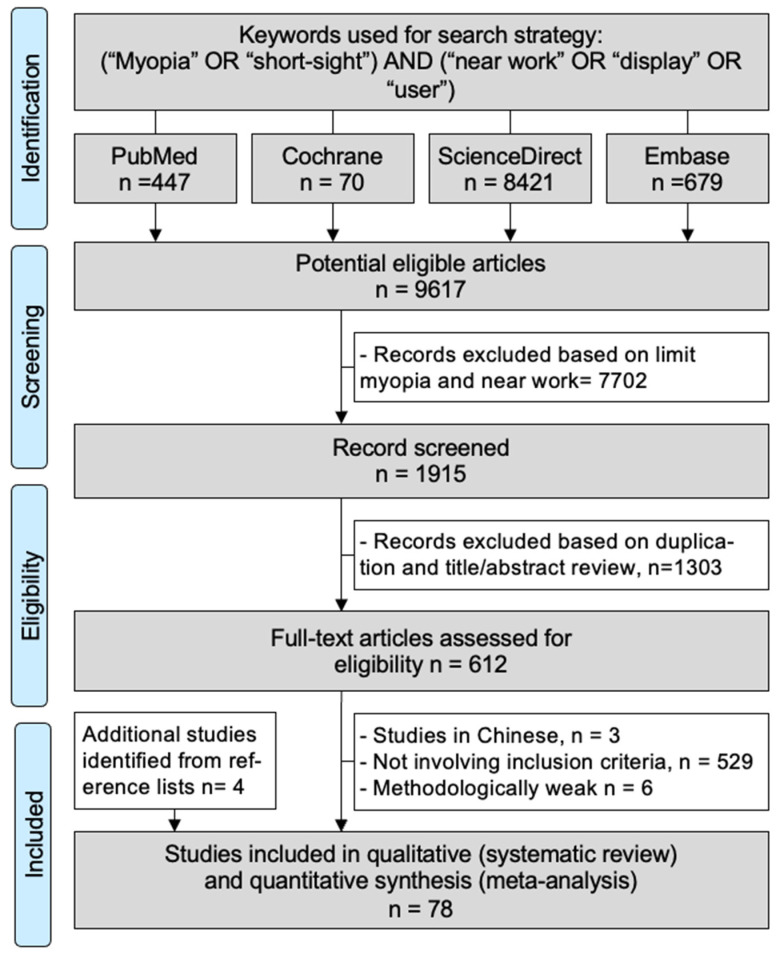
Flow chart.

**Figure 2 ijerph-20-00875-f002:**
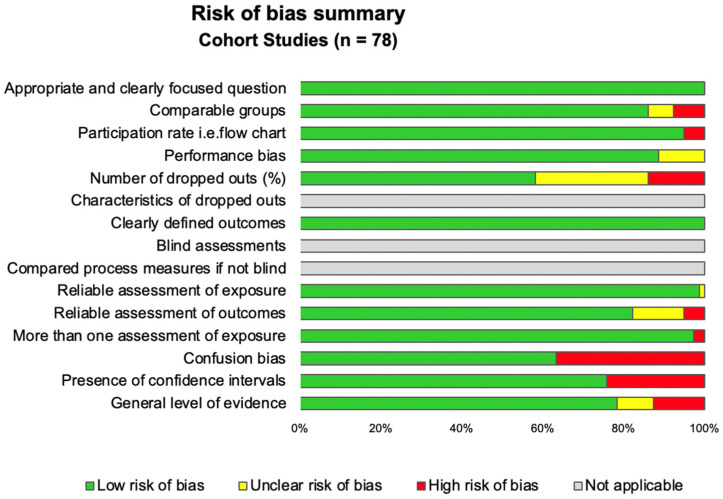
Summary of risk of bias of the included articles using the SIGN checklist.

**Figure 3 ijerph-20-00875-f003:**
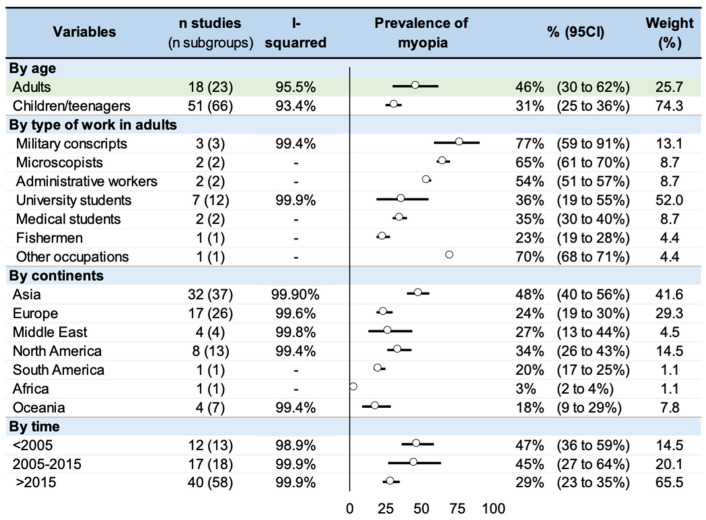
Summary of meta-analysis on the prevalence of myopia.

**Figure 4 ijerph-20-00875-f004:**
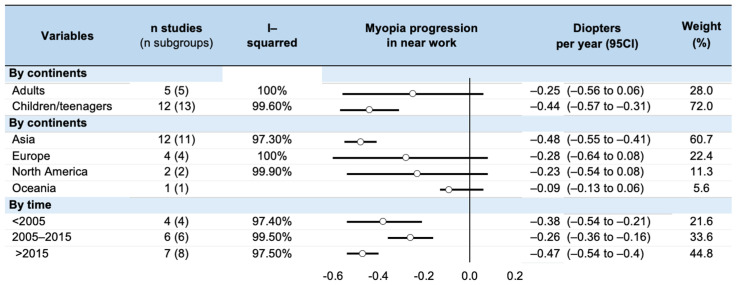
Summary of meta-analysis on the myopia progression per year in near work.

**Figure 5 ijerph-20-00875-f005:**
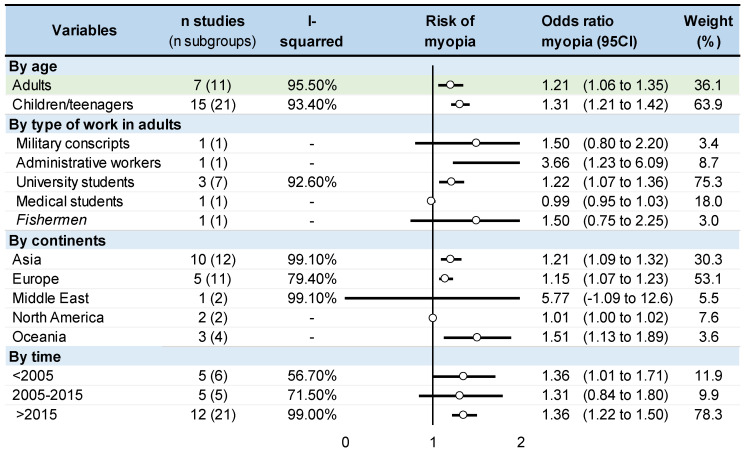
Summary of meta-analysis on the odds ratio for risk of myopia in near work.

**Figure 6 ijerph-20-00875-f006:**
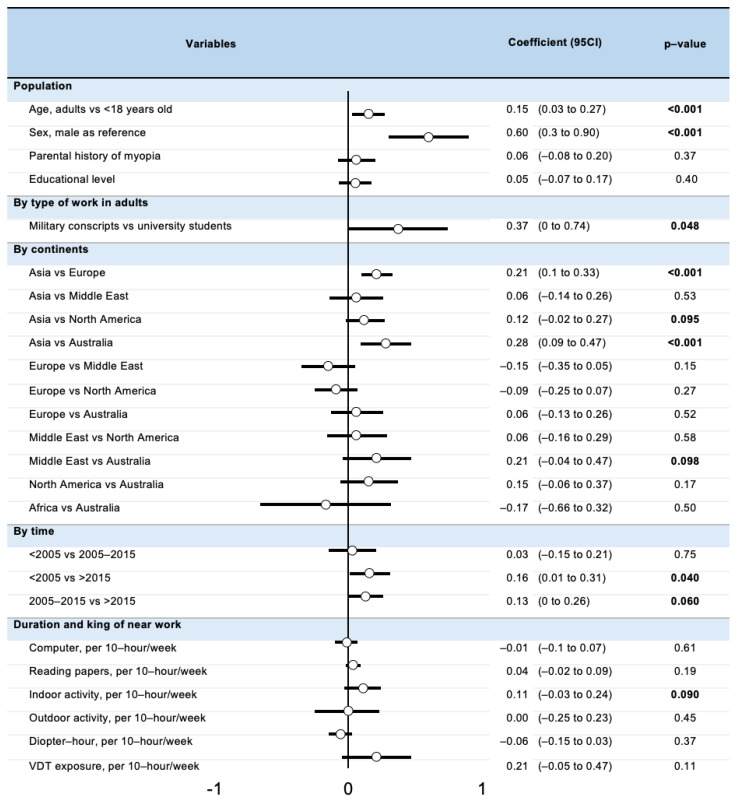
Metaregressions on factors influencing the prevalence of myopia in near work.

**Table 1 ijerph-20-00875-t001:** Characteristics of included studies. * Not included in the meta-analysis for prevalence.

Study	Country	Study Design	Characteristics of Population						Myopia			Type of
			n	Age, Years	Sex,% men	Children/ Adolescent	Adults	Occupation	Prevalence	Progression	Risks	Near-Work
Adams 1992 [26]	England	Cross-sectional	251	29.7	35.8		X	Microscopist	X	X		Diopter-hour
Alomair 2021 [27]	Saudi Arabia	Cross-sectional	850	10.5	55.9	X			X		X	unspecified
Alsaif 2019 [28]	Saudi Arabia	Cross-sectional	338	unknown	48		X	University student	X	X		unspecified
Atowa 2020 [29]	Nigeria	Cross-sectional	1197	11.5	45	X			X	X		Read, computer, video games, TV
Basso 2006 [30]	Italy	Cross-sectional	209	39.2	53		X	Administrative workers	X	X	X	VDT
Berhane 2022 [31]	Ethiopia	Cross-sectional	484	22.81	63.4		X	University student	X		X	Reading, computer
Bez 2019 [32]	Israel	Cross-sectional	22,823	17.7	100	X			X		X	unspecified
Chiang 2020 [33]	USA	Cross-sectional	6571	15.39	50.98	X			X		X	TV, computer
Czepita 2010 [19]	Poland	Cross-sectional	5865	11.9	48	X			X	X		Reading, computer, TV
Demir 2021 [79]	Sweden	Cross-sectional	128	12	45.3	X			X			unspecified
Deng 2010 [35]	USA	Prospective	147	12	54.4	X			X		X	Reading, TV, computer
Donovan 2012 [36]	China	Cross-sectional	85	10.3	51	X			*	X		unspecified
Edwards 1999 [37]	China	Prospective	123	7	57	X			X	X		VDT
Enthoven 2020 [38]	Netherland	Cross-sectional	8563	7	unknown	X			X		X	computer, reading
Enthoven 2021 [39]	Netherland	Cross-sectional	525	13.7	46	X			X			smartphone
Fernandez-Montero 2015 [40]	Spain	Prospective	6963	38.5	31		X	University student	X	X	X	Computer
French 2013 [41]	Australia	Prospective	4118	16	50	X			X		X	Diopter-hour
Giloyan 2017 [42]	Armenia	Cross-sectional	1092	12.95	46.6	X			X		X	unspecified
Guo 2013 [44]	China	Cross-sectional	681	12.95	unknown	X			*		X	Reading, computer
Guo 2016 [43]	China	Cross-sectional	3055	13.60	52	X			X		X	computer, reading
Guo 2017 [45]	China	Prospective	382	6.30	unknown	X			X	X	X	Reading, TV, computer
Han 2019 [46]	Korea	Cross-sectional	3398	36.3	54.5		X	unknown	X		X	Unspecified
Hansen 2020 [47]	Denmark	Cross-sectional	1443	16.6	45	X			X		X	Electronic device
Hinterlong 2019 [48]	Taiwan	Cross-sectional	3686	11.2	52	X			X		X	unspecified
Holton 2021 [49]	Taiwan	Cross-sectional	6200	11.2	53	X			X		X	unspecified
Hsu 2017 [34]	Taiwan	Prospective	3526	7.49	56	X			*	X	X	computer, unspecified
Huang 2019 [50]	China	Cross-sectional	968	19.6	66		X	University student	X		X	Computer, electronic device,
Huang 2020 [51]	Taiwan	Prospective	10,743	10	52.73	X			X	X	X	unspecified
Hung 2020 [96]	Vietnam	Cross-sectional	1987	14	50.3	X			X		X	Read, computer, TV
Ip 2008 [70]	Australia	Cross-sectional	2339	12.7	unknown	X			X		X	reading
Jacobsen 2008 [52]	Denmark	Prospective	143	23.1	unknown		X	Medical student	X	X		unspecified
Jones 2007 [53]	USA	Prospective	514	8	unknown	X					X	TV, computer, reading, diopter-hour
Jones-Jordan 2011 [14]	USA	Prospective	1318	10	48	X			X			TV, computer, reading
Jones-Jordan 2012 [54]	USA	Prospective	835	10.4	43	X			*	X		unspecified
Khader 2006 [55]	Jordan	Cross-sectional	1777	14.5	61	X			X		X	reading, TV, Computer
Kim 2020 [56]	Korea	Cross-sectional	938	12.2	51.4	X			X		X	unspecified
Kinge 2000 [97]	Norway	Prospective	192	20.6	48		X	University student	*	X	X	reading
Konstantopoulos 2008 [57]	Greece	Cross-sectional	200	21	100			Military conscript	X			Computer, reading, TV
Ku 2019 [58]	Taiwan	Prospective	1956	unknown	49.5	X			X	X	X	Computer, reading
Lam 1999 [59]	China	Prospective	142	11.38	47	X			X	X		unspecified
Lanca 2022 [60]	China	Cross-sectional	12241	9	52	X			X		X	TV, reading,
Lee 2013 [15]	Taiwan	Cross-sectional	5048	21.4	100		X	Military conscript	X			TV, reading, computer
Lee 2017 [61]	China	Prospective	23,114	unknown	52	X			X		X	Reading, TV, computer
Li 2015 [62]	China	Cross-sectional	1770	12.7	48	X			X			Reading, computer, TV, electronic device
Lin 2014 [63]	China	Cross-sectional	370	unknown	47	X			X			Reading, computer, TV
Lin 2016 [64]	China	Cross-sectional	836	10.5	52	X			X		X	unspecified
Lin 2016 [95]	China	Prospective	222	10.9	48.2	X			X	X	X	Indoor activities
Lin 2017 [65]	China	Cross-sectional	572	10.6	49	X			X		X	Reading, TV
Liu 2021 [66]	China	Cross-sectional	3831	unknown	51.5	X			X		X	Digital screen time
Loman 2002 [67]	USA	Cross-sectional	177	27	58		X	University student	X		X	Non specified
Lu 2009 [68]	China	Cross-sectional	998	14.6	44	X			X		X	Reading, TV
Ma et al. 2018 [69]	China	Prospective	1639	8.1	51.5	X			X	X	X	reading
Mavrakanas 2000 [71]	Greece	Cross-sectional	1738	unknown	unknown	X			X			Reading
McBrien 1997 [72]	England	Prospective	251	29.7	36		X	microscopist	X	X		Microscopy use
McCrann 2021 [73]	Ireland	Cross-sectional	402	16.7	45	X			X		X	Electric device
Muhamedagic 2014 [74]	Bosnia	Prospective	100	21.89	39		X	University student	*	X		Tv, computer, reading, diopter-hours
Mutti 2002 [75]	USA	Cross-sectional	366	13.7	55	X			X		X	Reading, TV, computer
Onal 2007 [76]	Turkey	Prospective	207	20.3	55		X	Medical student	X	X		unspecified
Pärssinen 2014 [13]	Finland	Prospective	240	10.9	50	X			X	X	X	Reading, TV
Pärssinen 2022 [77]	Finland	Cross-sectional	13,649	11	14.5	X			X		X	Reading
Patel 2019 [78]	India	Cross-sectional	248	17.8	55	X	X	Medical student	X			Tv, computer, electronic device, reading
Qi 2019 [80]	China	Prospective	522	15.5	100	X				X	X	Reading
Rose 2008 [81]	Australia	Cross-sectional	1735	9.7	51	X			X			TV, computer, Indoor activity
Saw 1999 [82]	Singapore	Prospective	405	9.5	unknown	X			*			Indoor activity
Saw 2001 [83]	China	Cross-sectional	429	21	100		X	military conscript	X		X	Indoor activity
Saw 2002 [84]	Singapore	Cross-sectional	1962	8	50	X			X		X	Indoor activity, reading
Speeg-Schatz 2001 [85]	France	Prospective	814	unknown	24		X	administrative worker	X			Diopter-hours
Sun 2018 [86]	China	Cross-sectional	4890	12.5	51.7	X			X		X	unspecified
Tsai 2016 [98]	Taiwan	Cross-sectional	11,590	unknown	53	X			X		X	Read, TV
Wong 1993 [2]	China	Cross-sectional	408	27	55		X	fisherman	X			TV, reading
Woodman 2011 [87]	Australia	Prospective	40	23.4	unknown		X	University student	X	X		unspecified
Wu 2010 [89]	Taiwan	Cross-sectional	145	9	52	X			X		X	indoor activities, computer, reading
Wu 2015 [88]	China	Cross-sectional	4677	16.9	47	X	X	University student	X		X	Reading
Yang 2018 [90]	Canada	Cross-sectional	166	9.6	50	X			X			unspecified
Yao 2019 [91]	China	Prospective	800	15	100	X			*	X	X	reading
Yotsokura 2020 [92]	Brazil	Cross-sectional	421	10.6	unknown	X			X			unspecified
You 2016 [93]	China	Prospective	4129	8.11	54	X			*	X	X	Reading, TV, indoor activities
Zadnik 2015 [94]	USA	Cross-sectional	4512	9.2	50	X			X			unspecified

## Data Availability

All relevant data are within the paper and its Appendix A.

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
