# Peer review of "Myopia and Near Work: A Systematic Review and Meta-Analysis"

_ijerph, 2023, doi:10.3390/ijerph20010875_

Round 1

Reviewer 1 Report

The Author of the Article has done exact meta-analysis of the publications  of myopia.  Author analized every environmental factors which has influence on myopia and its progression. 

The work title is well defined and results are summary of lot of publicated papers. The title of the paper is well defined and the results are a summary of many published papers.

The results are properly interpreted and the conclusions summarized the results.

The analyzes were made very carefully and based on a modern statistical workshop.

However, this Article may be of interest to a limited number of people due to its character.

Author Response

[REPLY] Dear Reviewer, we thank you very much for your very positive and salient comments. Considering that there are approximately 4 billion people currently working worldwide, and considering the pandemic of myopia, we believe that our article may interest many readers and may attract many citations.

Reviewer 2 Report

The work in my opinion is prepared in a correct manner. Congratulates the authors on the idea and execution.  I have minor suggestions that should be improved before publication.

1.      Introduction - In my opinion, the introduction should be expanded to include the following points:

a.      information that most often myopia is associated with axial elongation of the eyeball (10.1111/opo.12812 , 10.1159/000317072 ).

b.     possible sequelae of myopia (e.g., glaucoma, macular degeneration, retinal detachment, cataracts) (10.1097/IAE.0000000000001489 , 10.1167/iovs.61.4.49 ).

c.    information that myopia is also associated with changes in the musculo-fascial system (10.3390/jpm12040626 , PMID: 16646640 ).

2. Discussion – ‘’visual function [1] myopia can lead to fatigue, ‘’ –‘’[1]’’ - Is this citation correct?  The form of citation differs from the others.

3. The references is not prepared according to the style of the journal.

Author Response

[REPLY] Dear Reviewer, we thank you for your comments and suggestions. We have modified the Introduction to include the points that you suggested, to give a more thorough overview of the problem of myopia in the introduction.

In the Introduction section, we have amended the following sections to include your insightful suggestions:

  • « Myopia is the most common type of refractive error (1) and constitutes a rising global health issue causing significant impact on visual function (2,3) with potential consequences on quality of life, work productivity (4,5) and other sequelae (glaucoma, macular degeneration, retinal detachment and cataract) [REF : Ikuno, Haarman]. Myopia most often involves the axial elongation of the eyeball [REF: Chamberlain, Meng], and (…) »,
  • « Myopia progression is associated with long-term changes in the musculo-fascial system [REF: Zielinski] and could be associated with prolonged near work and less time spent performing sports and outdoor activities among children and adults (14). »

We have also modified the incorrectly typed reference in the Discussion, as suggested.

We thank you for pointing the misalignment of the reference style with the journal requirements. We have corrected it.

We hope this modified version of the manuscript will meet your expectations.

Reviewer 3 Report

This is a systematic review and meta-analysis looking at the association between near work and myopia. The authors found that the odds of myopia in workers exposed vs. non-exposed to near work were increased by 26% (18 to 34%), by 31% (21 to 42%) in children and 21% (6 to 35%) in adults. This paper was quite clearly carefully completed and written, but I see one major flaw. Given that these studies are not experimental design studies, there is a fairly high risk of confounding. The main confounder here, mostly not measured, is exposure to outdoor light. So, the authors are stating that near work is the main risk factor for myopia. But what could be happening is that outdoor light exposure is associated with myopia (supported by the scientific literature)  and outdoor light exposure is associated with near work exposure (presumably less near work with increasing outdoor exposure). At the very least, this possibility needs to be addressed and discussed in the discussion. I would add more qualifying language in the conclusion to soften it. I don't agree with " Near work conditions, including occupational exposure in adults, are associated with myopia." This needs to be softened. If possible, I would reassess the analysis adjusting for outdoor light exposure.

Author Response

[REPLY] Dear Reviewer, we thank you for your comments and suggestions. Please find hereafter our response:

  • Regarding the risk of counfounding factors, i.e. outdoor light exposure: Due to the observational nature of most of the studies included, and the lack of reporting of outdoor light exposure in some articles, the results regarding outdoor light exposure may be biased, but our results do not suggest an association between outdoor light exposure and myopia progression. We have discussed this potential confounding factor in the discussion, as you suggested.
    • We have modified the “Limitations” section as follows: « One potential confounding factor could be the amount of exposure to outdoor light one is exposed to, a parameter not reported in every included study. We could hypothesize that outdoor light exposure may be inversely correlated with indoor light exposure and near work, and thus could also be associated with myopia, but our results do not support this hypothesis (Figure 6). »
  • We considered with interest your suggestion regarding the conclusion. We have amended the manuscript to include more uncertainty in our conclusion of:
    • the abstract
      • From : " Near work conditions, including occupational exposure in adults, are associated with myopia."
      • To : " Near work conditions, including occupational exposure in adults, could be associated with myopia."
    • The manuscript :
      • From: “The main findings were that near work exposure, including occupational exposure in adults, is associated with myopia.”
      • To: “The main findings were that near work exposure, including occupational exposure in adults, could be associated with myopia.”

We hope this modified version of the manuscript will meet your expectations.

Reviewer 4 Report

The paper here presented is fine and welcome to myopia literature. It seems to be a very well done search of the literature both in children and in young adults. This has importance as myopia is developed mainly in schoolchildren in urban high incidence Asian environments, but shares a great adult onset incidence in other parts of the world with lower prevalence… So the study suggests more studies in adults and this idea is welcome.

Introduction and Methods are really fine. No changes suggested there except that the number of reviewers is “the first 2 raters” in line 90 page 2 if I am correct.

The results are very well presented in the text and tables, but the title “Results” is missing in my copy and seems to begin with a Table… I suggest to paste some of the text before beginning with a table. And I also suggest that the data for outdoor exposure revealed in those studies is presented in a table by continents and occupation as it is very well known that Asian children spend less time outdoors and it would be nice to see time outdoors versus occupation.

My principal concerns come with the Discussion.

After the publication of a paper on the genetics of myopia in 2005 by Ian Morgan 1 it became very clear that the epidemic in Asia is not genetic and the finding of more than 400 genes with very low penetrance each in myopia development2 shows that any subject or any population is prone to myopia if the environmental effects of intense nearwork and low outdoor exposure are at work. So it cannot be said that the epidemic in Asia could be genetic. That part should be deleted to my knowledge. And possibly discussed in the context of higher near work and less outdoor exposure in Asian children. If you have data about this go ahead and present them. If not cite the literature, please.3-7

It is very interesting that you have found that VDTs and book reading influence myopia both the same and it is interesting that you discuss this in the context of VDTs emitting blue light that could be protective.8,9

The idea that accommodation during reading produces myopia development has been probably left away by most researchers. Now the main reasons for the link between myopia development and reading have to do with “accommodative lag” and “contrast issues”…. I will explain myself. Every child and young adult lags a mean of -0.75 D while reading. And this posterior pole hyperopic defocus sustained for longs period per day promotes eye growth according to the theory proposed as early as 1993.10 This idea has now been confirmed as peripheral myopic defocus spectacles and contact lenses have consistently shown that myopia progression is slower when this potent myopic defocus is added at the posterior pole of the eye while reading. 11-13 On the other hand, the contrast issue was recently discovered by the team of Frank Schaeffel in Tubingen. He showed clearly that the simple fact of reading black text under white background (as usual) was one of the reasons for the association of myopia and reading. He showed both opposite choroidal changes in humans while reading with black or inverted contrast (white letters in dark background). He also showed the effect on ocular growth and myopia development in chicks exposed to both opposite contrasts in the visual environment. 14,15 His experiments are underway and only a trial putting myopic children to read under both opposite contrasts is need for further proof. But many practitioners are now recommending to read and use cellphones with inverted contrast.16

So taking this in account I expect changes in the Discussion when this paper is revised.

1.            Morgan I, Rose K. How genetic is school myopia? Progress in retinal and eye research 2005; 24: 1-38.

2.            Tedja MS, Haarman AEG, Meester-Smoor MA, Kaprio J, Mackey DA, Guggenheim JA, Hammond CJ, Verhoeven VJM, Klaver CCW. IMI - Myopia Genetics Report. Investigative ophthalmology & visual science 2019; 60: M89-M105.

3.            Morgan IG, He M, Rose KA. EPIDEMIC OF PATHOLOGIC MYOPIA: What Can Laboratory Studies and Epidemiology Tell Us? Retina 2017; 37: 989-97.

4.            Morgan IG, French AN, Rose KA. Intense schooling linked to myopia. BMJ 2018; 361: k2248.

5.            Morgan IG WP, Ostrin LA, et al. IMI Report on risk factors for myopia: From associations to causal mechanisms and preventive interventions. Investigative ophthalmology & visual science 2021; in press.

6.            Wu PC, Chen CT, Chang LC, Niu YZ, Chen ML, Liao LL, Rose K, Morgan IG. Increased Time Outdoors Is Followed by Reversal of the Long-Term Trend to Reduced Visual Acuity in Taiwan Primary School Students. Ophthalmology 2020; 127: 1462-9.

7.            Iribarren L IR. Myopia and Culture. Journal of Clinical and Experimental Ophthalmology 2022; 13: 1-5.

8.            Jiang X, Pardue MT, Mori K, Ikeda SI, Torii H, D'Souza S, Lang RA, Kurihara T, Tsubota K. Violet light suppresses lens-induced myopia via neuropsin (OPN5) in mice. Proceedings of the National Academy of Sciences of the United States of America 2021; 118.

9.            Mori K, Torii H, Hara Y, Hara M, Yotsukura E, Hanyuda A, Negishi K, Kurihara T, Tsubota K. Effect of Violet Light-Transmitting Eyeglasses on Axial Elongation in Myopic Children: A Randomized Controlled Trial. Journal of clinical medicine 2021; 10.

10.          Gwiazda J, Thorn F, Bauer J, Held R. Myopic children show insufficient accommodative response to blur. Investigative ophthalmology & visual science 1993; 34: 690-4.

11.          Anstice NS, Phillips JR. Effect of dual-focus soft contact lens wear on axial myopia progression in children. Ophthalmology 2011; 118: 1152-61.

12.          Lam CSY, Tang WC, Tse DY, Lee RPK, Chun RKM, Hasegawa K, Qi H, Hatanaka T, To CH. Defocus Incorporated Multiple Segments (DIMS) spectacle lenses slow myopia progression: a 2-year randomised clinical trial. The British journal of ophthalmology 2020; 104: 363-8.

13.          Bao J, Huang Y, Li X, Yang A, Zhou F, Wu J, Wang C, Li Y, Lim EW, Spiegel DP, Drobe B, Chen H. Spectacle Lenses With Aspherical Lenslets for Myopia Control vs Single-Vision Spectacle Lenses: A Randomized Clinical Trial. JAMA ophthalmology 2022; 140: 472-8.

14.          Aleman AC, Wang M, Schaeffel F. Reading and Myopia: Contrast Polarity Matters. Scientific reports 2018; 8: 10840.

15.          Wang M, Aleman AC, Schaeffel F. Probing the Potency of Artificial Dynamic ON or OFF Stimuli to Inhibit Myopia Development. Investigative ophthalmology & visual science 2019; 60: 2599-611.

16.          Galan MM SA, Fernandez Irigaray L, Rodriguez G, Aguirre R, Kotlik C, Iribarren R. Consenso de Miopia. Oftalmologia Clinica y Experimental 2022.

Author Response

[REPLY] Dear Reviewer, we thank you for those insightful suggestions. Please find hereafter our modifications based on your suggestions :

  • Methods - number of reviewers is “the first 2 raters” in line 90 page 2 :
    • We have corrected this incorrect typo. The corrected sentence now includes “the first 2 raters”.
  • Concerning the title “Results”: The title Results appears in Page 3, and begins by the paragraph “quality of articles”. We will check that this title appears in the correct place, long before the first table in the copy you will receive.
  • Regarding the presentation of outdoor light exposure, we have analysed the data of outdoor light exposure in the articles where it was reported (cf. Figure 6), without finding an association between outdoor light and myopia progression. We have discussed this potential confounding factor in the Limitations section, as follows :
    • « One potential confounding factor could be the amount of exposure to outdoor light one is exposed to, a parameter not reported in every included study. We could hypothesize that outdoor light exposure may be inversely correlated with indoor light exposure and near work, and thus could also be associated with myopia, but out results do not support this hypothesis (Figure 6). »

  • Regarding your comment on the potential role of genetics in myopia, we have taken into account your wise suggestions.
    • The last sentence of the “ Effect of time and influence of geographic zones” section has been modified as follows : “The role of genetic factors in geographic variations have been discussed, especially in East-Asia (106). Nevertheless, a large number of genes found to be potentially linked with myopia with low penetrance [REF: Morgan 2005, Tedja] suggests a common predisposition towards myopia in the environments with higher amount of near work and lower outdoor light exposure [REF: Morgan 2017, Morgan 2018, Morgan 2021, Wu 2020], a feature especially frequent in Asia. »
  • We have taken into account your suggestions regarding the accommodation and contrast.
    • We have incorporated those 2 issues and the literature that support them in the “Myopia progression in near work and dose response relationships” section, as follows:
      • According to some studies, myopia is significatively associated with continuous reading, longer duration of reading and shorter reading distance (46,94) [REF: Gwiazda J, 1993], due to the promotion of eye growth by sustained accommodation. Moreover, myopia may also be linked to the issue of contrast between support and text, as suggested by Aleman et al. [REF: Aleman 2018] and Wang et al. . [REF: Wang 2019]. One controversial hypothesis is that computer work might prevent myopia thanks to the release of dopamine induced by the brightness of digital screens, which could prevent eye growth (i.e. axial length), and therefore myopia (117).

We hope this modified version of the manuscript will meet your expectations.